# Is There an Association between Paw Preference and Emotionality in Pet Dogs?

**DOI:** 10.3390/ani12091153

**Published:** 2022-04-29

**Authors:** Tim Simon, Elisa Frasnelli, Kun Guo, Anjuli Barber, Anna Wilkinson, Daniel S. Mills

**Affiliations:** 1Department of Life Sciences, University of Lincoln, Lincoln LN6 7DL, UK; elisa.frasnelli@unitn.it (E.F.); abarber@lincoln.ac.uk (A.B.); awilkinson@lincoln.ac.uk (A.W.); dmills@lincoln.ac.uk (D.S.M.); 2CIMeC—Center for Mind/Brain Sciences, University of Trento, Piazza della Manifattura 1, 38068 Rovereto, TN, Italy; 3Department of Psychology, University of Lincoln, Lincoln LN6 7DL, UK; kguo@lincoln.ac.uk

**Keywords:** dog, emotionality, emotional predisposition, laterality, locomotion, motor bias, paw preference

## Abstract

**Simple Summary:**

Similar to human handedness, many animals preferentially use one limb over the other to perform certain tasks. It has been argued this handedness/limb preference could be linked to individual differences in emotionality (i.e., the tendency to appraise the environment in certain ways emotionally). A better understanding of this under-explored link has the potential to establish limb preference as an indicator of emotional vulnerability and increased risk for affective disorders (e.g., anxiety disorder, depression). This study explored a potential relationship between paw preference and emotionality in pet dogs. To determine paw preference, we examined paw preferences to hold a food-stuffed Kong™ and to lift first when starting walking or when stepping over a hurdle. Dogs’ emotionality was assessed using a validated owner-based questionnaire examining their sensitivity to positive and negative situations. In contrast to previous research this study did not find any association between dogs’ paw use and their emotionality but suggested that paw use might be task-specific. More research is needed to identify which limb is actually the preferred one in a given task and to define both the consistency and nature of any bias before drawing conclusions about their relationship with emotional appraisal of the environment.

**Abstract:**

Research with humans and other animals has suggested that preferential limb use is linked to emotionality. A better understanding of this still under-explored area has the potential to establish limb preference as a marker of emotional vulnerability and risk for affective disorders. This study explored the potential relationship between paw preference and emotionality in pet dogs. We examined which paw the dogs preferentially used to hold a Kong™ and to perform two different locomotion tests. Dogs’ emotionality was assessed using a validated psychometric test (the Positive and Negative Activation Scale—PANAS). Significant positive correlations were found for dogs’ paw use between the different locomotion tasks, suggesting that dogs may show a more general paw preference that is stable across different types of locomotion. In comparison, the correlations between the Kong™ Test and locomotion tests were only partially significant, likely due to potential limitations of the Kong™ Test and/or test-specific biomechanical requirements. No significant correlations were identified between paw preference tests and PANAS scores. These results are in contrast to previous reports of an association between dog paw preference and emotionality; animal limb preference might be task-specific and have variable task-consistency, which raises methodological questions about the use of paw preference as a marker for emotional functioning.

## 1. Introduction

Many animals show functional limb preferences whereby one limb is used in preference to the other to perform a particular task [1]. However, such preferences vary across taxa and populations. A meta-analysis by Ströckens et al. [1] revealed that of the 119 vertebrate species considered, 20 species showed evidence of limb preference at an individual level (i.e., in single individuals, regardless of any common directional bias in the population), while 61 species showed limb preferences at a population level (i.e., most individuals of a population displayed a bias in the same direction). Amongst vertebrates, the most prominent example of limb preference is human handedness, with ~90% of the population showing a right-hand preference in activities such as writing or fine-tuned object manipulation [2]. There is also mounting evidence of preferential limb use at an individual or population level in a range of invertebrates [3].

The domestic dog (*Canis familiaris*) is an increasingly popular model species for the study of limb bias, with several studies reporting on paw preference and a variety of tests used to determine preferential paw use. The most widely used test is the Kong™ Test, which identifies the paw a dog preferentially uses to hold a Kong™ (KONG Company, Golden, CO, USA), a hollow, conical-shaped toy, while retrieving food placed inside it [4,5,6,7,8,9,10,11]. Other tests record, for instance, which paw is preferentially used to reach for food in a container or under a sofa [12,13,14], to remove an adhesive tape or blanket from the dog’s head [12,15,16,17], to “Give Paw” on request [12,18], or to start walking from a stationary position [10,18,19]. According to a recent meta-analysis, paw preference is a common feature, with 68% of dogs showing either a left or right paw bias at the individual level, but with no directional bias at the population-level [20].

Research on humans and other animals, including dogs, has suggested a link between limb preference and emotional functioning, indicating that individual differences in limb preference are related to variation in emotionality (i.e., the tendency to appraise the environment emotionally in certain ways and to show emotional responses of certain types) and varying risks of developing mental disorders associated with emotional functioning. However, the mechanisms underlying this link remain unclear [21,22]. A better understanding of the relationship between limb preference and emotional functioning has the potential to establish limb preference as a valuable marker of emotional vulnerability and increased risk for certain disorders, with potential implications for animal husbandry and handling, the identification of at-risk groups, and possibly even for psychotherapy strategies [23]. Generally, given that both limb preference and emotional functioning are related to asymmetries between the left and right brain hemisphere, it can be hypothesized that individual differences in limb preference might reflect differential brain asymmetry patterns that also extend to networks relevant to emotional processes and variation in emotional functioning as a result.

Although the relationship between limb preference and brain structure and function remains to be elucidated in dogs, limb preference does seem to be associated with differential asymmetric brain organization in other species. In humans, for instance, movements of the dominant hand lead to relatively higher activity patterns in the contralateral brain hemisphere, while movements of the non-dominant hand are associated with more bilateral activity [24,25]. In both humans and rodents, limb preference is linked to neuroanatomical and neurochemical left–right differences in different networks of the brain [26,27,28,29,30,31,32].

Specific emotional functioning has also been associated with asymmetries between the two brain hemispheres in humans and other animals [21,33,34,35,36,37,38]. However, the precise role of each hemisphere remains unclear. Different competing hypotheses have been proposed; The Right Hemisphere Hypothesis, for example, claims that right-hemispheric networks are generally more specialized for emotional functioning than left-hemispheric networks [33,34], whereas other hypotheses maintain that networks of each hemisphere may play dominant roles depending on circumstances. For instance, the Valence Hypothesis maintains that left- and right-hemispheric networks are generally more specialized in response to stimuli with a positive and negative emotional valence, respectively [35]. If individual differences in limb preference are associated with varying patterns of structural and/or functional hemispheric asymmetries in networks involved in emotional processes, such asymmetry differences could correspond with an increased/decreased dominance of the hemisphere specialized for emotional functioning in the respective networks, leading to inter-individual variation in emotional functioning.

In humans, an association between limb preference and emotional functioning has been suggested, for instance, by research reporting that human handedness may be linked to anxiety [39,40,41] and mental disorders associated with emotional functioning, such as depression [42,43,44,45] or post-traumatic stress disorder (PTSD) [42,46,47,48].

In several non-human animals, individual differences in preferential limb use has also been associated with variation in emotional functioning (e.g., non-human primates [49], elk [50], and cats [51]). In dogs, ambilateral individuals have been reported to show more fear-related behavioral responses when attending to presumably alarming sounds (thunderstorms and fireworks), compared to left- or right-paw preferent dogs [4]. Differences in dogs’ paw preference have also been associated with performance in behavioral evaluations. For example, using the Canine Behavioural Assessment & Research Questionnaire (C-BARQ; [52]), Schneider et al. [53] found that left-/right-pawed dogs yielded higher scores for stranger-directed aggression, compared to ambilateral individuals. This might suggest that dogs with a paw preference had a relatively higher emotional predisposition to appraise strangers as potentially threatening and to show aggressive behavior accordingly. This finding could not, however, be reproduced by a more recent study that reported higher scores for stranger-directed aggression in right-pawed and ambilateral dogs, compared to left-pawed individuals [11]. A study on shelter dogs revealed a positive correlation between increasing left-pawedness and the frequency of changes in body position, the display of vocalizations, and lower body postures, indicating a higher vulnerability to stress and negative affective states [54]. This inconsistency in findings raise question over the comparability of the methods chosen as well as more fundamental questions about the nature of limb laterality and emotionality.

All these dog studies employed the Kong™ Test to measure paw preference. Although the Kong™ Test is widely used, its interpretation and validity as a measure of paw preference is unclear. First, given that feeding behavior is associated with a general specialization of left-brain networks, which indirectly manifests itself in different lateralized behaviors in various species [55,56,57], dogs’ motor response in the Kong™ Test could be biased by food-related lateralization patterns. Therefore, the Kong™ Test may be less indicative of a general paw preference, but rather it may measure a highly task-specific motor laterality pattern that reflects food-related specialized hemispherical functioning. Second, concerns have been raised as to whether the paw used by dogs to hold the Kong™ is actually their dominant one [58]. An alternative interpretation of the Kong™ Test suggests that the paw on the floor that provides postural support during the task is actually the dominant paw [58]. Finally, dogs’ performance in the Kong™ Test might be influenced by inadvertent learning. If a dog is more successful in retrieving food using a particular paw to stabilize the Kong™, this will increase the probability of that same paw being used again. However, purely by coincidence, a dog could learn that using its non-dominant paw is more successful. Similar unwanted learning effects might also play a role in other paw preference tests assessing, for instance, which paw is preferentially used to reach food in a container/under a sofa, to remove a plaster/blanket from the dogs’ head, or to “Give Paw” on command. It should also be noted that giving a paw on request usually involves a behavior that many owners will have taught their dog for a reward and may be subject to the owner’s own hand bias [59]. Given the limitations of these tests, other tests should be considered to measure paw preference, especially when considering a potential link between paw preference and emotional functioning. For instance, spontaneous locomotion tests, such as assessing which paw is preferentially used to start walking from a stationary position, might be a more suitable alternative. Given that paw use in such locomotion tests is unlikely to be biased by task-specific behavioral lateralization patterns due to other specialized hemispherical functions, locomotion tests may be a preferable indicator of general paw preference in dogs. This assumes that locomotion tests measure spontaneous, unconditioned behavior that is not reinforced by rewards, and so they should be less susceptible to unwanted learning effects.

In this context, Barnard et al. [10] explored a possible link between paw preference and dogs’ personality by using, in addition to the Kong™ Test, a locomotion test that assessed which paw the dog preferentially lifts first in order to walk down a step. Some of the personality traits assessed (measured by the Dog Mentality Assessment (DMA) scores; [60]) were likely associated with emotional predispositions. For instance, when paw preference was assessed with the Kong™ Test, higher scores for aggressiveness in response to threatening stimuli and playfulness were found in ambilateral dogs, compared to left-/right-pawed individuals. However, these results were not consistent with the outcomes of preference from the locomotion test: no association between paw preference and aggressiveness or playfulness was identified. Despite this, an association between paw preference and the sociability of dogs was reported, with individuals showing ambilateral paw use obtaining higher scores for this trait than left-/right-pawed individuals. Sociability was based on dogs’ greeting behavior towards strangers, their willingness to make close physical contact, and to cooperate with them. Thus, sociability scores might be a more subtle measure of dogs’ emotional predisposition towards strangers.

The aim of the current experimental study was to examine the consistency of the relationship between paw preference and emotional functioning in dogs, with this species used as a model species for consideration of wider issues in this field of research. Paw preference was assessed in terms of absolute preference accounting for left versus right bias and strength of preference (i.e., lateralization of response regardless of direction). Assuming that a general limb preference is reflected in preferential limb use that is consistent across different motor tasks [61], and provided that (some) individual dogs show a general paw preference, it was hypothesized that the performance of paw-preferent dogs should positively correlate between different locomotion tests, without necessarily being positively correlated with their performance in the Kong™ Test. Dogs’ emotional predispositions were examined using a validated psychometric test (the Positive and Negative Activation Scale—PANAS; [62]) which assesses sensitivity to rewards and aversives.

## 2. Materials and Methods

### 2.1. Subjects

Pet dogs of various breeds were recruited by approaching owners via social media channels (e.g., Facebook, Twitter). As this study involved the investigation of whether dogs prefer to use one or the other fore paw when executing certain motor tasks, only dogs were included that, according to their owners’ judgement, did not suffer from medical conditions which could affect their performance (e.g., musculo-skeletal problems or impaired vision). Data were collected from a total of 86 dogs (55 purebred dogs, 31 mongrels; 37 females—28 neutered, 49 males—34 neutered; age (mean ± SD): 5.5 ± 3.2 years; details of dogs are provided in Appendix A).

### 2.2. Experimental Procedure

#### 2.2.1. Paw Preference Tests

Dog’s paw use was assessed in three different tests conducted in the dog’s home. Dog owners did not have to carry out all the tests. The order in which the tests were asked to be executed was randomized for each dog. Written instructions and demo videos of how to do the tests were provided (Appendix A). Dog owners were advised to split the tests over at least 2–3 days and had 14 days to complete the study after they had started it. Dog owners sent test scores to a secure server belonging to the University of Lincoln (UK) (for details see Appendix A).

##### Kong™ Test

This test was based on that used by Tomkins et al. [19] and assessed paw preference by recording how often a dog used their left or right paw to hold a Kong™, (Figure 1a, KONG Company, Golden, CO, USA), when retrieving food from it. The size of Kong™ used depended on the dog’s breed and weight, and recommendations of the KONG Company (https://www.kongcompany.com/de/size-guide-by-breed, accessed on 29 April 2022). Before the test, the dog owner filled the Kong™ with moist dog food. The test was carried out when dog owners would normally feed their dog; instead of/before having their regular meal, the dog received the food-stuffed Kong™. The test started after the dog had laid down and the Kong™ was placed in front of the dog so that it was equidistant from both front paws. For the starting position, it was important that the dog’s body was evenly and symmetrically aligned, with a straight spine and both hind and front limbs parallel to each other (Figure 1b). During the test, the dog was free to move.

Following Tomkins et al. [19], the paw used to stabilize the Kong™ on each of 50 occasions was recorded. Each individual observation was recorded only after the dog removed their paw(s) from the Kong™. If both paws were used to hold the Kong™, the interaction was recorded, but not included in the 50 unilateral paw uses. Figure 2 shows an example of a dog holding the Kong™ with either the left, right or both paws. To avoid the test taking too long, the Kong™ could be removed from the dog’s possession if they stabilized the Kong™ with their paw(s) for more than 2 min; the Kong™ was then placed back in front of the dog as previously described. The dog owner observed the dog and recorded the paw uses using a tally list. If the dog retrieved all the food from the Kong™ before 50 paw uses were recorded, a newly filled Kong™ was provided to complete the test.

##### First Step Test

This test assessed paw preference by recording the first front paw a dog used to start walking from a stationary stand (Condition 1), sit (Condition 2), or lying position (Condition 3). For the stationary start position of each condition, it was important that the dog’s body was evenly and symmetrically aligned, with a straight spine and both hind and front limbs parallel to each other (Figure 3). After the dog was in the start position, the dog owner squatted down in a straight line in front of their dog (distance: approximately 2 m). The dog owner called their dog and noted which paw the dog lifted first to start walking. A total of 5 trials were carried out for each start position.

##### Hurdle Test

This test assessed paw preference by recording which paw a dog used first to step over a small hurdle. Depending on the items available in the dog owner’s home, the hurdle could be built, for example, using a broom or a similar length stick and books, bowls or cooking pots (Figure 4). To control for varying dog sizes, the height of the hurdle should be about half the length of the dog’s front leg (see an example in Figure 4). The Hurdle Test included the same three different stationary start positions as the First Step Test; dogs stepped over the hurdle from a stand (Condition 1), a sit (Condition 2), or a lying position (Condition 3). For the start position of each condition, the distance between the dog and the hurdle should be such that the dog could place their two front paws directly over the hurdle without needing additional intermediate steps forward. After the dog had taken up the respective start position, the dog owner squatted down in a straight line in front of their dog (distance: approximately 2 m). As in the First Step Test, the dog owner called their dog and noted which paw the dog used to lead over the hurdle first. The Hurdle Test was carried out in order of the above-mentioned start positions (Conditions 1–3), with a total of 5 trials being performed for each start position.

#### 2.2.2. PANAS Questionnaire

Dog owners were asked to complete the Positive and Negative Activation Scale (PANAS), a 21-item questionnaire that assesses emotionally negative (i.e., frequency of fearful and relaxed states, responses to changing and unfamiliar environments, habituation, and the startle response) and positive (i.e., energy and interest, persistence, and excitement) activations in dogs (https://ipstore.lincoln.ac.uk/product/the-positive--negative-activation-scale-panas-for-dogs, accessed on 29 April 2022; [62]). Higher negative activation scores indicate a tendency to assess the environment as potentially threatening and to develop fears or phobias, whereas higher positive activation scores are indicative of a predisposition to perceive the world as rewarding or appealing and to pro-actively engage with it [62].

### 2.3. Data Analysis

All data were analyzed using R Studio 4.0.5 [63].

#### 2.3.1. Paw Preference

To quantify each dog’s paw preference, a *directional laterality index* (LI) was calculated for each test and condition. The LI was calculated by (L − R)/(L + R), where L indicates the number of times the left paw was used, and R the number of times the right paw was used. The LI thus quantifies paw preferences on a scale from −1 (complete right paw preference) to +1 (complete left paw preference), where 0 means that the animal showed no preference [4,12]. In addition to the directional LI, the *degree of paw preference*, i.e., regardless of the direction, was considered an alternative dependent variable. This was determined from the absolute value of their LI scores (ABS–LI) for each test and condition. Thus, the ABS–LI measures the strength of paw preference on a scale from 0 (no lateral preference) to +1 (strong lateral preference).

For each test and condition, it was investigated whether the dogs showed any directional paw preference by testing if the dogs’ LI scores deviated significantly from 0 using one sample Wilcoxon signed-rank tests. Bonferroni corrections were applied to control for multiple testing and results are reported after this correction. It was further investigated whether the degree of lateralization (i.e., preference regardless of direction) differed from 0 in the different tests and conditions using the same statistical testing procedure on the ABS–LI data.

To test whether dogs’ paw preference varied between different conditions within the First Step or the Hurdle Test, the LI scores of the different conditions were compared by employing Friedman rank sum tests. To further explore the relationship of the LI data between the conditions within the First Step or Hurdle Test, Spearman’s correlations tests were carried out between the pairwise combinations. Bonferroni corrections were applied to control for multiple correlation tests. The same procedure was applied using the ABS–LI scores for the different conditions. In the absence of significant differences between the conditions within the First Step and/or Hurdle Test conditions, it was deemed acceptable to pool scores from the conditions to generate pooled LI and ABS–LI values. It was then assessed whether dogs’ pooled scores deviated from 0 using one sample Wilcoxon signed-rank tests.

To assess whether dogs’ paw preference varied between the Kong™ Test, the pooled LI scores were compared using a Friedman rank sum test. Spearman’s correlations tests were used for additional exploration of the pairwise relationships within the LI data between the three tests. This procedure was then repeated for the ABS–LI data.

#### 2.3.2. Correlation between Paw Preferences and PANAS Scores

PANAS scores for positive and negative activation were calculated as per the authors’ recommendations [61]. To test whether paw preference was associated with differences in emotional predisposition, Spearman’s rank correlation coefficients were calculated to describe the relationship between dogs’ LI data for each test (i.e., Kong™ Test and the pooled First Step and Hurdle Test) and dogs’ PANAS scores. Bonferroni corrections were applied to control for multiple comparisons. To test whether the strength of paw preference was associated with differences in emotional predisposition, the same procedure was undertaken using dogs’ ABS–LI data.

## 3. Results

### 3.1. Paw Preference Tests

In total, 23 of the 86 dogs that participated in the study completed the Kong™ Test. Additionally, 82 dogs participated in the First Step Test; 75 of these 82 dogs performed this test for all three conditions, two individuals performed the test for Conditions 1 + 2, two individuals did Conditions 1 + 3, two dogs did Conditions 2 + 3, and one dog did the test only for Condition 1. Furthermore, 48 dogs participated in the Hurdle Test; 47 of these dogs completed the test for all three conditions and one dog performed the test for Conditions 1 + 2. Nineteen individuals carried out all tests and conditions (i.e., Kong™ Test and all conditions of both the First Step and Hurdle Test), and 44 dogs performed all conditions of both the First Step and the Hurdle Test. One dog executed the Kong™ Test and all conditions of the First Step Test, and two dogs completed the Kong™ Test and all conditions of the Hurdle Test. All of the 86 dogs that carried out the different tests and conditions were included in the data analyses.

The dogs’ LI scores did not differ significantly from 0 for any test or condition (all *p* ≥ 0.294), except for Condition 3 of the First Step Test (V = 1003, *n* = 79, *p* = 0.049) (Figure 5). The dogs’ ABS–LI scores differed from 0 for all tests and conditions (all *p* < 0.001) (Figure 6).

The dogs’ LI scores did not significantly differ between conditions for either the First Step (Friedman χ^2^ = 2.469, df = 2, *n* = 75, *p* = 0.291) nor the Hurdle Test (Friedman χ^2^ = 0.964, df = 2, *n* = 47, *p* = 0.618). Moderately positive correlations were found between all conditions within the First Step (0.461 ≤ Spearman’s Rho ≤ 0.513, all *p* = 0.006; Table 1) and the Hurdle Test (0.383 ≤ Spearman’s Rho ≤ 0.434, all *p* = 0.018; Table 1). For neither the First Step (Friedman χ^2^ = 4.739, df = 2, *n* = 75, *p* = 0.094) nor the Hurdle Test (Friedman χ^2^ = 3.262, df = 2, *n* = 47, *p* = 0.196) did the dogs’ ABS–LI scores significantly differ between conditions. No significant correlations between conditions within the First Step or Hurdle Test were found (all *p* ≥ 0.15; Table 1), except for a weakly positively correlation between Condition 2 and 3 of the First Step Test (Spearman’s Rho = 0.297, *n* = 78, *p* = 0.048; Table 1).

The LI scores significantly deviated from 0 for the pooled First Step Test (V = 1149, *n* = 82, *p* = 0.016) but not for the Kong™ Test (V = 174.5, *n* = 23, *p* = 0.294) nor for the pooled Hurdle Test (V = 479, *n* = 48, *p* = 0.264) (Figure 7). The ABS–LI scores differed significantly from 0 for the Kong™ Test (V = 231, *n* = 23, *p* < 0.001) and both pooled tests (all *p* < 0.001) (Figure 8).

There was a tendency towards the LI scores differing between the Kong™, the pooled First Step, and the pooled Hurdle Test, but it did not quite meet the threshold for statistical significance (Friedman χ^2^ = 5.6, df = 2, *n* = 19, *p* = 0.061). Moderately positive correlations were detected both between LI scores of the Kong™ and the pooled Hurdle Test (Spearman’s Rho: 0.478, *n* = 20, *p* = 0.033; Table 2) and the pooled First Step and the pooled Hurdle Test (Spearman’s Rho = 0.478, *n* = 46, *p* = 0.006; Table 2); no correlation was observed between LI scores of the Kong™ and the First Step Test (Spearman’s Rho: 0.275, *n* = 19, *p* = 0.255; Table 2). The ABS–LI scores did not vary between the Kong™, the pooled First Step, and the pooled Hurdle Test (Friedman χ^2^ = 2.179, df = 2, *n* = 19, *p* = 0.336). A moderately positive correlation was found between the LI scores of the pooled First Step and the pooled Hurdle Test (Spearman’s Rho: 0.354, *n* = 46, *p* = 0.017; Table 2); no correlations were found between LI scores of other tests (all *p* ≥ 0.697; Table 2)

### 3.2. Correlation between Paw Preferences and PANAS Scores

PANAS scores were obtained for 84 of the 86 dogs that carried out paw preference tests. The median scores for positive and negative activation were 0.68 (IQR: 0.188) and 0.438 (IQR: 0.255), respectively. For none of the tests (i.e., Kong™ Test, pooled First Step Test, pooled Hurdle Test) was there a significant correlation between either LI or ABS–LI scores and PANAS scores (all *p* ≥ 0.186; Table 3).

## 4. Discussion

In addition to the widely used Kong™ Test, the current study employed different locomotion tests to assess paw preference in dogs and explored a potential association between preferential paw use and dog emotionality. While a directional population-level paw preference was only detected in the context of the First Step Test, a population-level bias in paw use, i.e., regardless of the direction, was found for all tests and conditions. Overall, dogs’ directional paw preference was more consistent across different tests and conditions, compared to the strength of their paw preference (regardless of direction). Dogs’ directional paw preference was consistent across different locomotion tasks, but correlated only partially between locomotion tests and the Kong™ Test. No correlations were found between measures of paw preference and PANAS scores. These results raise general methodological questions about the assessment of paw preference in dogs and challenge the previously proposed use of paw preference as a marker of emotionality in dogs.

### 4.1. Paw/Limb Preference

#### 4.1.1. Direction, Strength, and Generality of Paw/Limb Preference

The dogs’ LI data suggest a right-biased population-level paw preference in the context of the First Step Test (i.e., for both Condition 3 and the pooled test), but no directional population-bias for the Hurdle Test (i.e., neither for any particular condition nor for the pooled test) or the Kong™ Test. This finding corresponds with previous studies reporting no directional population bias for the Kong™ Test [10,18,19,54] and either a right-lateralized [19] or no bias [10,18] when paw preference was measured by a locomotion test recording which paw a dog preferentially uses first to walk down a step.

The dogs’ ABS–LI data indicate a population-level bias in paw use (regardless of the direction) for all tests and conditions. Similar results were provided when paw preference was assessed by determining the paw a dog preferentially used first to walk down a step [10,18]. However, previous findings regarding ABS–LI data for the Kong™ Test are mixed; some are in line with the outcome of the current study [10,18], whereas other studies found that dogs’ ABS–LI did not significantly deviate from 0, indicating a lack of population-level bias and, thus, ambilaterality [19,54].

For both the LI and the ABS–LI data, no significant differences could be detected between the different conditions within particular tests, nor between the Kong™ Test, the pooled First Step, and the pooled Hurdle Test. However, while LI scores were significantly correlated between all conditions within particular tests, between the pooled First Step and the Hurdle Test, and between the Kong™ and the pooled Hurdle Test, the ABS–LI scores were only significantly correlated between two conditions of the First Step Test and between the pooled First Step and the pooled Hurdle Test. As the LI appears as the more consistent measure across different motor tasks, compared to the ABS–LI data, future studies should especially focus on dogs’ directional LI as a measure of lateralized paw use.

Assuming that a general limb preference should be reflected in a preferred limb use that is consistent across different motor tasks [61], the consistency of the dogs’ LI scores in the context of the First Step and Hurdle Test suggests that dogs may indeed have a more general paw preference that is reflected in different locomotion behavior. However, considering that the LI scores were only moderately correlated and that the dogs’ ABS–LI scores were only significantly correlated between two conditions of the First Step Test and between the pooled First Step and the pooled Hurdle Test, the current study also indicates that preferential limb use in locomotion tasks is not an invariable, but rather a dynamic behavior. As this was the first study that compared dogs’ motor responses in different locomotion tasks, more work is needed to explore the stability of limb preference in different locomotion contexts.

#### 4.1.2. Which Paw/Limb Is the Dominant One?

Compared to locomotion tests, concerns have been raised about the adequacy of the Kong™ Test as a potential indicator of a limb preference. Therefore, it was not expected that the dogs’ results in the pooled First Step and pooled Hurdle Test would necessarily correlate with those of the Kong™ Test. Whereas no correlation was observed between the Kong™ and the pooled First Step Test, a significant positive correlation was detected for dogs’ LI scores between the Kong™ and the pooled Hurdle Test. Considering that the Kong™ Test might potentially be biased by a left-hemispheric specialization for feeding contexts and inadvertent learning, the validity of the Kong™ Test as a measure of paw preference can be questioned; this might account for why the Kong™ Test was only partially correlated with the locomotion tests. An alternative explanation could be that potentially different and test-specific biomechanical requirements led to a significant correlation between the Kong™ and the Hurdle Test, but not between the Kong™ and the First Step Test. Some authors have questioned whether the paw used by dogs to hold the Kong™ is actually their dominant one [58]. An alternative interpretation of the Kong™ Test suggests that the paw on the floor that provides postural support during the task is actually the dominant paw [58] or some other aspect of laterality may be occurring. Holding the Kong™ with one paw typically coincides with the dog’s head being turned to the contralateral side so that the dog can retrieve the food from the Kong™ with their mouth (Figure 2). This leads to an asymmetrical distribution of the body weight which is typically counterbalanced by the paw that is static on the ground. Possibly, providing the required postural support is more demanding than holding the Kong™ and, thus, the paw which is more frequently used to stabilize the dogs’ body position could be considered as the dominant paw; alternatively, the direction of head turn might be a more appropriate measure of motor laterality. This challenges the assumptions of what is lateralized in the subject.

In addition, compared to the First Step Test, an increased demand for positional control may also be involved in the Hurdle Test. Assuming that the dog is required to lift their leading paw both higher and for a prolonged time when stepping over a hurdle, the Hurdle Test involves an increased asymmetric body weight distribution, compared to the First Step Test, and thus requires more postural support of the contralateral forelimb on the floor. As a result, with increasing demands for positional control, the dominant paw might be used differently: When the demands for position control are rather low, such as in the First Step Test, the dominant paw might be preferred to be lifted first to start walking. Yet, when the requirement for compensating an asymmetric body weight distribution is higher, such as in the Hurdle Test, the dominant paw might be increasingly used to stabilize the body, allowing the non-dominant paw to move over the hurdle first. On the one hand, the motor action pattern of the Hurdle Test is sufficiently similar to the motor action pattern in the First Step Test, accounting for the correlation of both test outcomes. On the other hand, higher demands for body stabilization in the Hurdle Test, compared to the First Step Test, might induce a functional shift of some dogs’ dominant paw towards an increased use for positional control, explaining the correlation of dogs’ performance between the Kong™ and the Hurdle Test. However, even if it is assumed that the paw that provides most postural support during the Hurdle Test is the dominant one, the data of the current study are not sufficient to determine with certainty which paw this is. The proposed hypothesis that the paw which initially remains on the floor engages more in stabilizing the body position and, thus, qualifies as the dominant paw, needs to be tested by future studies. This hypothesis can only be retained if the postural support of the initially static paw is greater overall than the postural support that is provided by the contralateral paw (i.e., the paw that is lifted first to step over the hurdle) when it contacts the ground again behind the hurdle. To determine which paw provides more positional control during the Hurdle Test, future investigations should record both paws’ contact times with the ground and estimate force differences between both paws (e.g., by using force plates). Future studies on lateralized paw use in dogs should not only focus on dynamic movements, but also consider the static use of paws to provide postural support and positional control.

An alternative explanation as to why the Kong™ Test was not significantly correlated with both locomotion tests could be that the Kong™ Test measures paw preference less validly than the locomotion tests. Specifically, the Kong™ Test might potentially be biased by a left-hemispheric specialization for feeding contexts and/or inadvertent learning effects.

#### 4.1.3. Paw/Limb Preference or Functionally Differentiated Use of Both Paws/Limbs?

Previous studies and, thus far, also the current one, have tended to discuss lateralized paw use in dogs within the conceptual framework of paw preference and in terms of the dichotomous notion of the dominant vs. the non-dominant paw. However, considering that some tests, such as the Kong™ or the Hurdle Test, involve both dynamic movements of one paw and static use of the contralateral paw to satisfy increasing demands for positional control, the question arises as to how appropriate the concept of paw preference/paw dominance is at all with regard to such tests. Rather than identifying *the preferred paw*, some tests might introduce a dual-task, whereby *each paw is differentially used* to perform one of the two exercises involved. That is, some tests might reveal a complementary and functionally differentiated use of both paws, rather than an undifferentiated preference of a single paw. In the Kong™ Test, for instance, one paw might be preferentially used to hold the Kong™, a task that is presumably associated with more fine-tuned motor actions and object manipulation, whereas the other paw is more likely to be used for postural support, requiring both strength and stamina. Conceivably, the Hurdle Test is associated with a similar complementary use of both paws, with one paw being preferentially used to perform more precise and fine-tuned movements when leading over the hurdle, and the other paw being preferentially used for position control. A similar functional differentiation that also involves both fine-tuned motion and position controlling behavior has been described for dogs in the context of galloping. When galloping, dogs (and other quadrupeds) display a lateralized cyclic motor pattern, whereby each limb plays a unique role [64]. In each cycle of a galloping sequence, a trailing and a leading forelimb can be distinguished, with the trailing limb contacting the ground first, followed by the leading limb. The trailing limb is associated with greater accelerating forces and more precise and fine-tuned motion during the so-called pre-stance retraction, where the shoulder blade initiates a back rotation of the trailing limb before contact with the ground [65]. Compared to that, the leading limb has longer contact intervals with the ground and is associated with greater decelerating forces, thus providing more positional control [64]. The functional differentiation in gallop locomotion is associated with individual preferences of dogs to use one or the other limb more frequently during galloping as the trailing and leading limb, respectively [66].

Functionally differentiated limb use also plays a role in other animals. According to the “postural origins theory” proposed by MacNeilage et al. [67], hand preferences in primates, for instance, evolved from a postural position preference of the right hand and a left-hand preference for rapid strikes to seize small prey in insectivorous arboreal prosimians (early primates). In terrestrial species, where postural demands are less pronounced, the left hand remained the dominant one for stereotyped tasks such as reaching, whereas the right hand became the dominant one for more manipulative tasks [67]. A comparable specialization still exists in humans [67]; even right-handers are more accurate in pointing at a suddenly appearing small visual object when using their left hand [68]. Although right-handers display right hand advantage in accurate throwing, their left hand is superior in catching moving objects [67,69].

Future research on lateralized limb use in dogs and other species needs to be aware of the possibility that some motor behavior is indicative of a functionally differentiated use of a pair of limbs, rather than demonstrating the preferential use and undifferentiated dominance of a single limb.

### 4.2. Correlation between Paw/Limb Preference and PANAS Scores

Given that neither the LI data nor the ABS–LI data of any test were correlated with dogs’ PANAS scores, this study provides no evidence for a potential link between paw preference and emotional predisposition in dogs. This finding contrasts with prior studies that reported an association between dogs’ paw preference and their predisposition towards fear [4], emotionally motivated aggression [10,11,53], playfulness [10], sociability towards strangers [10], and their vulnerability to stress and negative affective states [54]. It should be noted that, unlike PANAS, the latter are all assessments of more extreme manifestations of emotionality.

In addition, the way laterality is defined may be important and vary between studies. Compared to the current study, some of the previously reported associations between paw preference and emotional disposition were not based on dogs’ LI or ABS–LI data but on a categorization of dogs as either right-pawed, left-pawed, or ambilateral. The three-way categorization may be based on binomial z-scores: using the formula $z=\frac{R-0.5N}{\surd 0.25N}$, where R signifies the number of times a dog used the right paw when performing a particular task and *N* signifies the sum of the dog’s left and right paw uses during that task; dogs with a z-score ≥ +1.96 were classified as right-pawed, those with a z-score ≤ −1.96 were typified as left-pawed, and the remainder were defined as ambilateral. Depending on whether dogs were classified into different laterality categories or whether their motor response was quantified in terms of LI and ABS–LI scores, previous studies came to different conclusions about the relationship between dogs’ paw preference and emotional functioning. For example, using the three-way categorization, the results reported by Wells et al. [11] suggested a generally higher predisposition towards stranger-directed aggression in dogs displaying either a right-paw or no lateral preference in the Kong™ Test. However, when dogs’ motor responses were quantified in terms of LI scores, the authors found that increasing right-pawedness was associated with relatively elevated scores for both stranger-directed aggression and fear, but only in a subgroup of dogs presenting with behavioral problems (assessed on the basis of C-BARQ scores). When Schneider et al. [53] based their analyses on a three-way categorization of lateralized paw use, they discovered that paw-preferent dogs showed higher scores for stranger-directed aggression, compared to ambilateral dogs. Yet, when using ABS–LI scores, they found no differences between more paw-preferent and more ambilateral individuals. Classifying dogs into laterality categories, Barnard et al. [10] found that ambilateral dogs displaying no paw preference in the Kong™ Test obtained higher scores for aggressiveness and playfulness. Moreover, assessing paw preference by recording the paw that was lifted first to walk down a step, dogs that were classified as ambilateral yielded higher scores for sociability towards strangers. However, when dogs’ motor responses were quantified in terms of LI and ABS–LI scores, the authors detected only a significant negative correlation between ABS–LI scores for the locomotion test and sociability, indicating that more ambilateral individuals scored higher in the sociability trait; no correlations were found between dogs’ paw use in the Kong™ Test and any personality traits that were related to emotional predisposition.

Although classifying dogs as either right-pawed, left-pawed, or ambilateral is a common practice in laterality research, analyzing dogs’ paw use in terms of the LI seems generally preferable (and the same may be said of emotionality too). First, given that the LI is a continuous variable containing information relating to both the direction and strength of lateralization, it provides more information than the three-way categorization, and thus offers more statistical power [70,71]. Second, grouping dogs into three categories appears somewhat arbitrary. One could just as well categorize dogs into fewer classes (e.g., two classes: right-pawed vs. left-pawed) or more classes (e.g., five classes: strongly right-pawed vs. moderately right-pawed vs. strongly left-pawed vs. moderately right-pawed vs. ambilateral). When dogs’ paw use is categorized, the used categories should be recognizable and distinguishable from each other when looking at the frequency distribution of individuals’ LI scores. Taking the frequency distribution of LI scores for human handedness, for instance, a typical “J”-shaped distribution can be identified [21] (p. 127), with most humans showing scores at one extreme of scale indicating right-handedness, some individuals displaying values at the other extreme of the scale indicating left-handedness, and only a few individuals showing LI scores close to 0 indicating ambilaterality. Such a “J”-shaped distribution would allow, for example, a discrimination between three different groups: e.g., right-handers vs. left-handers vs. ambilateral individuals. Compared to that, when looking at the frequency distributions of the dogs’ LI scores for the different tests and conditions used in the current study (Figure A1, Figure A2, Figure A3 and Figure A4), or the distributions presented in a previous study [18], comparable cut-offs distinguishing between different laterality classes cannot be identified.

### 4.3. Wider Implications

Even when only considering results based on dogs’ LI and ABS–LI data, the findings of previous work and those of the current study highlight potentially important inconsistencies. The current investigation could not replicate previous findings suggesting an association between individual limb preference and emotionality. This lack of replicability casts doubts about whether, or in what ways, limb preference can serve as a valid marker for emotional predisposition in dogs. However, as the potential link between preferential limb use and dogs’ emotionality is still under-explored, further research is needed to arrive at a more reliable conclusion about the presence or absence of such a link. Given that locomotion tests appear more appropriate than other tests that have been used to assess limb preference, future studies, in all species, should consider greater employment of locomotion tests. If the tests used to assess limb use might involve functionally differentiated use of both limbs, special attention needs to be paid as to which is selected to explore potential associations. These arguments and considerations apply regardless of the species being studied and may be especially important when considering comparative work.

### 4.4. Future Directions

Overall, the findings of our study seem to be robust. The lack of a population-level bias for dogs’ LI scores in Conditions 1 + 2 of the First Step Test and in the context of the Hurdle Test does not appear to be the result of sample size (minimum required sample sizes range between *n* = 180 and *n* > 4000) but rather a genuine lack of effect. However, a moderate increase in sample size (i.e., minimum required sample size: *n* = 44) may have seen an effect for the Kong™ Test. The absence of correlations between some paw preference tests and between some test conditions, and the lack of an association between the dogs’ performance in the paw preference tests and PANAS scores, do not appear to be the result of sample size either (the minimum sample sizes required range from *n* = 101 and *n* > 12,000), but rather indicate a lack of effect as well. However, a small increase in sample size (i.e., minimum required sample size: *n* = 33) may have revealed an effect between the Kong™ Test and the negative activation score, given the measured effect size.

This work suggests that locomotion tests may be better than the Kong™ Test to determine (a more general) paw preference in dogs as they, for example, do not involve food; this avoids a task-specific bias as a result of food-related dominant involvement of left-hemispheric networks. However, despite no food being used in the First Step and Hurdle Test, it cannot be entirely ruled out that some dogs nevertheless expected food or social reinforcement from the owner [72,73]. Both locomotion tests required the dogs to take up a stationary start position on command and finally to walk to their owner after being called. It is therefore possible that some dogs were expecting some form of reward for the completion of these tasks and that this expectation was associated with food-related and/or positive emotion-related hemispherically lateralized brain activity that could have influenced dogs’ motor response [31,32,33,34,35,36,37,54,55,56]. This is unlikely to account for our results, as they are not consistent between the Kong™ Test and the locomotion tests.

The current work used only a small number of trials for each condition in order to stimulate natural responses and to prevent both the dogs and their owners from being overburdened with the tasks. To gain more precise LI and ABS–LI scores, future work could potentially run trials in more controlled conditions or run more trials. All paw preference tests were carried out in the dogs’ home and laterality scores were exclusively based on their owners’ ratings. In future studies on paw preference that involve testing dogs in their home, dog owners could be asked to provide video recordings of the tests to assess the inter-rater reliability of behavioral analyses. Further investigations of paw/limb preference in dogs should not only focus on dynamic movements but also consider the static use of paws/limbs to provide postural support and positional control.

The current study used the PANAS questionnaire to assess dogs’ emotional predisposition. Although validated, this questionnaire provides an owner-based assessment of dog behavior that might be dependent on owners’ perception of situations and their general ability to read their dog’s behavior. It is therefore important to follow this work up with research that independently assesses dog emotionality using an ethogram. To date, research has focused on a (more general) paw/limb preference as a potential marker for emotional predisposition. An additional and complementary approach to identifying behavioral markers of emotionality could be to investigate whether lateralized motor behaviors (e.g., in the form of asymmetric paw/limb use or other motor behaviors) in response to particular emotionally relevant stimuli are linked to emotional predisposition. Research on different species [37,38], including dogs [17,38], indicates that motor responses towards emotionally relevant stimuli are lateralized. However, whether individual differences in the direction and/or strength of motor lateralization in specific emotional contexts correspond to variation in emotional predisposition is largely unexplored.

## 5. Conclusions

Even though the Kong™ Test is widely used, concerns can be raised about its appropriateness as a measure of paw preference. Therefore, different (presumably less problematic) locomotion tests were used as alternative additional measurements of paw preference. The consistency of dogs’ paw preference both between different conditions within particular locomotion tests and between different locomotion tests indicates that dogs might display a more general paw preference that is stable across different locomotion behavior. Compared to this, dogs’ paw preference correlated only partially between the Kong™ Test and locomotion tests. As the Kong™ Test might be influenced by a left-hemispheric specialization for feeding contexts and unwanted learning effects, limitations of the Kong™ Test might account for why the dogs’ performance in this test was only partially correlated with their results in the locomotion tests. An alternative explanation refers to varying demands for postural control and body stabilization, which might also explain why the results from Kong™ Test correlated with those from the Hurdle but not the First Step Test. To date, the role of postural control in dogs’ preferential paw use is largely unexplored.

Future studies, regardless of species, need to address this gap by specifically investigating biomechanical asymmetries between limbs, for instance measuring potential asymmetries in ground forces applied by the left and the right limb during tasks that are associated with an increased demand for body stabilization. Moreover, awareness of the possibility that some motor tasks are indicative of a functionally differentiated use of a pair of limbs, rather than demonstrating an undifferentiated dominance of a single limb, needs to be raised. This is particularly important when comparing lateralized limb use across different tasks and when considering potential relationships between limb use and other traits, such as emotionality.

## Figures and Tables

**Figure 1 animals-12-01153-f001:**
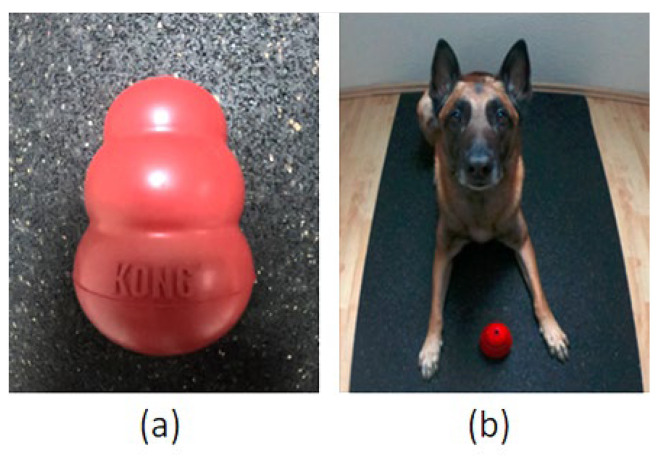
(**a**) A Kong™. (**b**) Start position for the Kong™ Test: The dog lies down with the dog’s body evenly and symmetrically aligned; the Kong™ is put between the dog’s front paws.

**Figure 2 animals-12-01153-f002:**
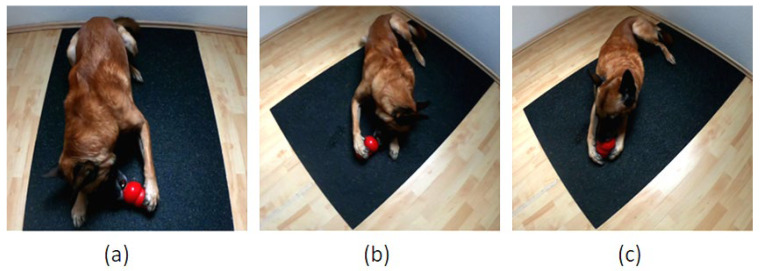
(**a**) Kong™ is stabilized with the left paw. (**b**) Kong™ is stabilized with the right paw. (**c**) Both paws stabilize the Kong™.

**Figure 3 animals-12-01153-f003:**
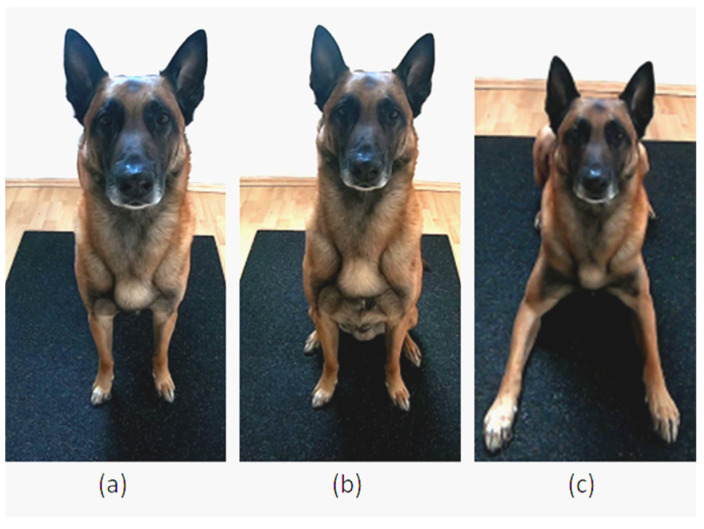
(**a**) Condition 1: Stand position. (**b**) Condition 2: Sit position. (**c**) Condition 3: Lying position.

**Figure 4 animals-12-01153-f004:**
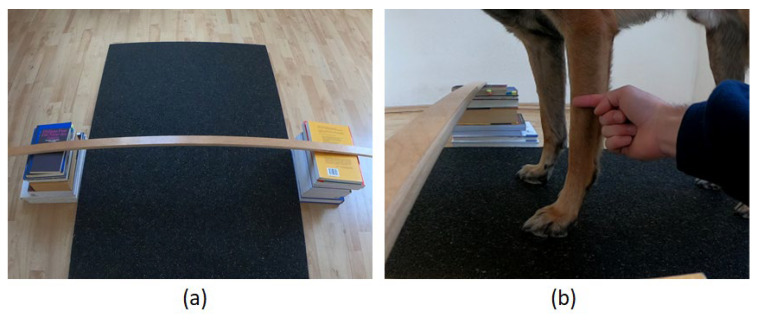
(**a**) Hurdle built from a slat and two piles of books. (**b**) The height of the hurdle measures half the length of the dog’s front leg.

**Figure 5 animals-12-01153-f005:**
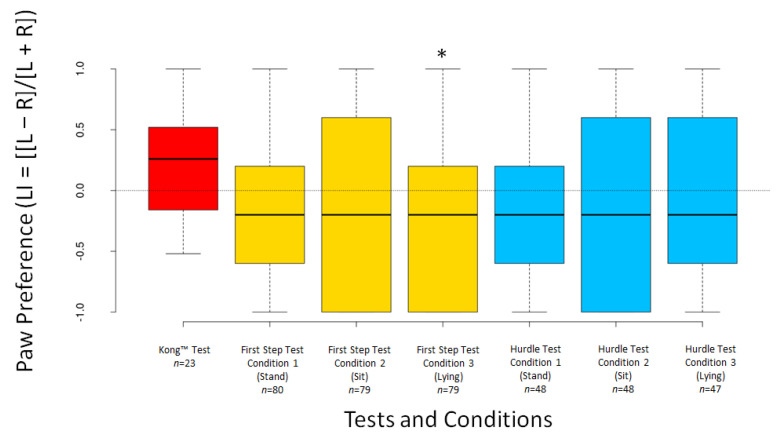
Boxplots showing dogs’ LI scores in the different tests and conditions. Asterisks indicate significant deviations from 0 (* *p* < 0.05).

**Figure 6 animals-12-01153-f006:**
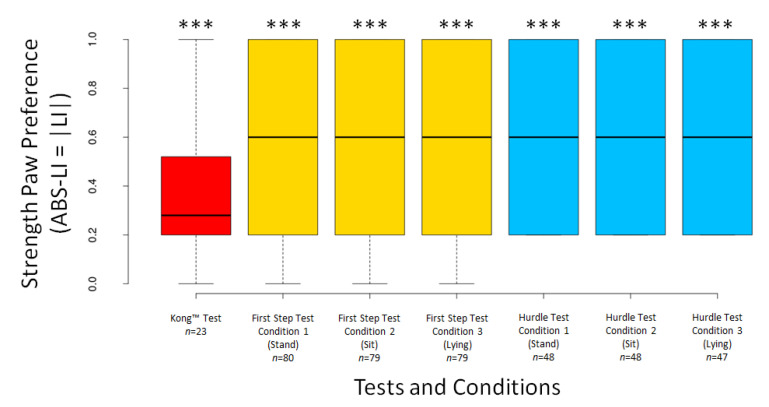
Boxplots showing dogs’ ABS–LI scores in the different tests and conditions. Asterisks indicate significant deviations from 0 (*** *p* <0.001).

**Figure 7 animals-12-01153-f007:**
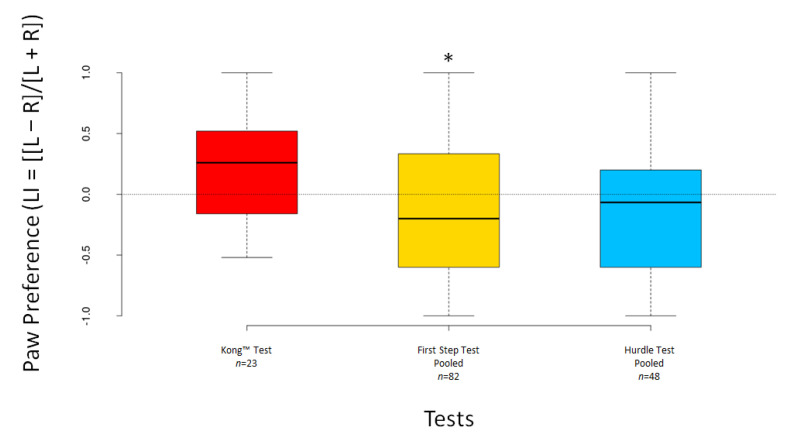
Boxplots showing dogs’ LI scores for the Kong™, the pooled First Step and the pooled Hurdle Test. Asterisks indicate significant deviations from 0 (* *p* < 0.05).

**Figure 8 animals-12-01153-f008:**
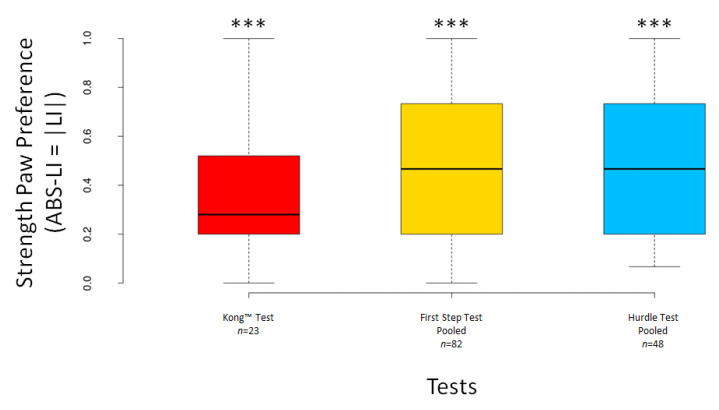
Boxplots showing dogs’ ABS–LI scores for the Kong™, the pooled First Step and the pooled Hurdle Test. Asterisks indicate significant deviations from 0 (*** *p* < 0.001).

**Table 1 animals-12-01153-t001:** Correlations of LI and ABS–LI scores between the different conditions within the First Step or Hurdle Test. Sample sizes (*N*), Spearman’s correlation coefficients (r_s_) and Bonferroni-corrected *p*-values are shown (* *p* < 0.05).

		LI		ABS–LI	
	*N*	r_s_	*p*-Value	r_s_	*p*-Value
First Step Test					
Condition 1~Condition 2	77	0.469	0.006 *	0.246	0.15
Condition 1~Condition 3	77	0.461	0.006 *	0.118	0.606
Condition 2~Condition 3	77	0.513	0.006 *	0.297	0.048 *
Hurdle Test					
Condition 1~Condition 2	48	0.408	0.018 *	0.243	0.309
Condition 1~Condition 3	47	0.383	0.018 *	0.123	0.606
Condition 2~Condition 3	47	0.434	0.018 *	0.292	0.196

**Table 2 animals-12-01153-t002:** Correlations of LI and ABS–LI scores between the different tests (i.e., Kong™ Test, pooled First Step and pooled Hurdle Test). Sample sizes (*N*), Spearman’s correlation coefficients (r_s_), and Bonferroni-corrected p-values are shown (* *p* < 0.05).

		LI		ABS–LI	
	*N*	r_s_	*p*-Value	r_s_	*p*-Value
Kong™~First Step	19	0.275	0.255	0.096	0.697
Kong™~Hurdle	20	0.478	0.033 *	0.025	0.918
First Step~Hurdle	46	0.404	0.006 *	0.354	0.017 *

**Table 3 animals-12-01153-t003:** Correlations between the LI and ABS–LI scores of the different paw preference tests (i.e., Kong™ Test, pooled First Step, and pooled Hurdle Test) and PANAS scores. Sample sizes (*N*), Spearman’s correlation coefficients (r_s_), and Bonferroni-corrected *p*-values are shown.

		LI		ABS–LI	
	*N*	r_s_	*p*-Value	r_s_	*p*-Value
Kong™~Positive Activation	21	0.083	0.768	0.15	1.000
Kong™~Negative Activation	21	0.471	0.186	0.132	1.000
First Step~Positive Activation	82	0.225	0.215	−0.041	1.000
First Step~Negative Activation	82	0.173	0.369	−0.209	0.372
Hurdle~Positive Activation	48	0.263	0.284	0.047	1.000
Hurdle~Negative Activation	48	0.129	0.768	0.166	1.000

## Data Availability

The data presented in this study are available in Appendix A.

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
