# Peer review of "Is There an Association between Paw Preference and Emotionality in Pet Dogs?"

_animals, 2022, doi:10.3390/ani12091153_

Round 1

Reviewer 1 Report

The authors carried out three experiments to test motor laterality in dogs (in the form of paw preference). Moreover, the potential relationship between paw preference and emotionality has been investigated. The manuscript is clearly written and well referenced. The theoretical considerations about the results are engaging.

I report below some important points that should be addressed and further analysed before publication.

My major concern regards the number of dogs participating to the study.

A description of the number of the subjects considered in the analysis is reported in the paragraph 3.1. but further details are needed. How many dogs performed each Conditions of the First Step and Hurdle test? How many of them were finally involved in the analysis? May the difference in the number of subjects included in each test have influenced the results?

How many subjects have been included for investigating the correlation between the Kong Test, the First Step and Hurdle test? Did all of them complete the three tests (and the related conditions)? It would be crucial to describe how powerful the results could be for a general understanding of the reliability of these tests. The N values for all the results need to be reported (also check the N reported in the Figure 5 and 6).

It seems reasonable to suppose that the observation regarding the correlation between motor tests have included 23 subjects (individuals that completed the Kong Test). If so, more carful considerations should be given to the results (and conclusions) about the consistency between tests.

Details regarding the subjects involved in the study should be included in the 2.1. paragraph (breeds, how many mixed breed dogs, age, sex -neutered or not).

As for the Hurdle test, details about the procedure should be provided for the First step test: Did the dog spontaneously start locomotion or was it under the owner command (calling the dog name)? Where was the owner positioned?

Were the dogs free to move during the Kong test or it was considered for the scoring only when the dogs laid down? In all the three tests the animal initial position was taken under owner oral-command (maybe to avoid any bias at the starting point). The need to maintain the position under command (for all the three tests) may have influenced dogs’ emotional state causing stress (or some expectation of reward?). This could have affected the paw preferential use (and dog spontaneous behaviour). This hypothesis needs to be discussed.

Regarding the evaluation of dog emotionality, authors reported some inconsistency in the results provided by questionnaires in recent studies. Although validated, the questionnaires provide owner-rated measures about dog behaviours that could be dependent on their perception of situations and their general ability to recognize dog emotional state. This could also explain the lack of correlation between motor laterality and emotionality here reported.

I find that the introduction needs to be more focused on the topic. For instance, I suggest removing unnecessary references to other species than dogs and the hypothesis on the hemisphere involvement for emotional processing (e.g. line 86-101, but also see 109-138….).

Line 70: reference lacking

Line 629: reference number in brackets (?)

Reviewer 2 Report

I appreciate the opportunity to review this document.
The research represents a contribution to ways to assess limb preference in dogs. However, the emphasis placed on results and discussion does not correspond to the title of the paper and its objective.

For the main result, they only report briefly in section 3.2 (L403-L408), it is necessary to add at least one table with all the results and deepen this analysis and the discussion about it.

Also, the method should be described in more detail. Authors reported that "Dog owners were advised to 226 split the tests over at least 2-3 days and had 14 days to complete the study after they had 227
started it. Results were sent to a secure server belonging to the University of Lincoln (UK)". Nevertheless, they do not say if participants sent records or videos, nither clarify if the videos were reviewed by a single person or if there were seveal, if authors calculated reliability between observers.

Round 2

Reviewer 1 Report

All the comments have been satisfactory addressed.

Author Response

According to Reviewer 1, all the comments have already been satisfactorily addressed.

Reviewer 2 Report

Authors responded satisfactorily to the comments received.
I consider that the paper is ready to be published

Author Response

According to Reviewer 2, all the comments have already been satisfactorily addressed and the manuscript is ready for publication.